# Reflecting on motivations: How reasons to publish affect research behaviour in astronomy

**Julia Heuritsch** *

Research Group "Reflexive Metrics", Institut für Sozialwissenschaften, Humboldt Universität zu Berlin, Berlin, Germany

* julia.heuritsch@hu-berlin.de, julia.heuritsch@gmail.com

## Abstract

Recent research in the field of reflexive metrics, which analyses the effects of the use of performance indicators on scientific conduct, has studied the emergence and consequences of evaluation gaps in science. The concept of evaluation gaps captures potential discrepancies between what researchers value about their research, in particular research quality, and what metrics measure. In the language of rational choice theory, an evaluation gap persists if motivational factors arising out of the internal component of an actor's situation are incongruent with those arising out of the external components. The aim of this research is therefore to study and compare autonomous and controlled motivations to become an astronomer, to do research in astronomy and to publish scientific papers. This study is based on a comprehensive quantitative survey of academic and non-academic astronomers worldwide with 3509 responses. By employing verified instruments to measure perceived publication pressure, distributive & procedural justice, overcommitment to work and observation of scientific misconduct, this paper also investigates how these different motivational factors affect research output and behaviour. I find evidence for an evaluation gap and that controlled motivational factors arising from evaluation procedures based on publication record drives up publication pressure, which, in turn, was found to increase the likelihood of perceived frequency of misbehaviour.

## 1. Introduction

*Reflexive metrics* is a strand in science studies, which explores how the demand for accountability and performance measurement in science has shaped the research culture in recent decades (e.g. [1,2]). Drawing on organizational culture theories (OCT) and rational choice theory (RCT), one can analyse how the prevailing research culture in turn shapes research behaviour (a variety of those studies and references can be found in [3]). For example, hyper-competition and publication pressure, which are shown to be part of the neoliberal research culture, may lead researchers to engage in scientific misconduct, compromising research integrity (e.g. [4,5]). Research integrity, in turn, has been linked to research quality (e.g. [6,7]).

**Data Availability Statement:** All relevant data are within the manuscript and its Supporting information files.

**Funding:** This study was performed in the framework of the junior research group "Reflexive

Metrics", which is funded by the BMBF (German Bundesministerium für Bildung und Forschung; project number: 01PQ17002). The funders had no role in study design, data collection and analysis, decision to publish, or preparation of the manuscript.

**Competing interests:** The authors have declared that no competing interests exist.

While outright misconduct, such as fabrication, falsification or plagiarism (FFPs) is rather rare, it may be the many more subtle questionable research practices (QRPs), which jeopardize the quality of the knowledge produced [3,8,9]. Concerns regarding research quality are raised and studied within the field of *reflexive metrics*.

This research is based on a quantitative survey among international astronomers. Astronomy is an interesting field to study from a meta perspective; it is dedicated to basic research and uses (open) archives and huge datasets. On the one hand, astronomy is an unusually collaborative field [10], on the other hand, pressures to perform on the individual level may inhibit knowledge sharing [11]. Astronomy is prone to performing research out of curiosity and nevertheless, the pressure to perform may lead to individualistic behaviours and scientific misconduct. Findings of studying astronomy from a reflexive metrics perspective may be generalizable to other sciences as well. Scientific fields, which promise more profitable output, such as (bio-) medicine, may experience similar pressures to astronomy, plus the added pressure from generating monetising applications.

A part of the data collected through this web-based survey was analysed by Heuritsch [12], who performed the first study bringing together the strand of research on the influence of the organisational culture on research integrity and *reflexive metrics*. Drawing on OCT, the author studied how role-associated factors (such as academic position and location of employment) and cultural factors (such as perceived publication pressure and distributive & procedural justice) affect research behaviour and quality in astronomy. According to the author's findings, is an astronomer who perceives publication pressure more likely to perceive work less rewarding, less procedural justice and the necessity to put more effort into the work. Moreover, the author finds that perceived publication pressure explains nearly 10% of the variance of observed misconduct.

According to RCT, not only the prevailing culture, including its (institutional) norms, but also the present material opportunities, and personal motivations comprise an *actor's situation* [13]. Institutional norms set the rules for what is appropriate behaviour and define what are desirable goals. Personal motivations are part of the so-called internal component of an actor's situation, while culture, norms and material opportunities make up the three external components. The actor's situation influences their decisions and hence their behaviour.

A qualitative study by Heuritsch [14], employing an RCT framework, found that astronomer's intrinsic motivation is to do research on the most fundamental questions of the universe out of curiosity and to "push knowledge forward". By contrast, their extrinsic motivation relates to complying with institutional norms, such as publishing a specific amount of papers per year, which is found to be the most important currency in the field. According to the author, this may lead to a perceived evaluation gap; a discrepancy between what researchers value about their research, in particular research quality, and what metrics measure. The author found evidence that astronomers hence experience an anomie; they want to follow their intrinsic motivation to pursue science in order to push knowledge forward, while at the same time following their extrinsic motivation to comply with institutional norms. This may result in a balance act to cope with the evaluation gap: the art of serving performance indicators in order to stay in academia, while at the same time compromising research quality as little as possible.

Building up on the qualitative research and quantitative research presented in [12,14], respectively, I study different motivations for becoming an astronomer, for doing research in astronomy and for publishing scientific papers resulting from that research. While the author [12] studied cultural and role-associated factors affecting the perception of misbehaviour occurrence, I aim at completing this rational choice analysis of the structural conditions in astronomy, by paying tribute to the motivational factors arising out of the astronomers'

research situation. I do so, by quantitatively comparing the autonomous motivation arising out of the internal component of an astronomer's situation with the controlled motivation arising out of the external components. In other words, I study how intrinsic and extrinsic motivations relate to each other. A discrepancy may quantitatively underpin the presence of the evaluation gap and resulting balance act found by the qualitative study in [14]. In order to complete this rational choice picture of the astronomer's research situation, I am bringing together the strands of reflexive metrics, RCT, OCT and self-determination theory (SDT).

This paper will be structured as follows: First, I will give a theoretical background on SDT's conceptualization of motivation and an account for the link between motivation and behaviour. Second, the method section describes the sample selection, the survey instruments, research question & hypotheses and technical aspects concerning the statistical analyses. Third, the result section contains the results from the EFAs, CFAs, descriptive statistics, and the regression models. The result section is followed by a discussion, strength & limitations and a conclusion section that also gives an outlook for future studies.

## 2. Theoretical background

### 2.1. The motivation continuum

This study's operationalisation of the concept of motivation is rooted in self-determination theory (SDT; [15–18]), which belongs to the domain of organisational psychology. SDT proposes a multidimensional conceptualization of motivation and offers explanations of how these different types of motivation can be en- or discouraged [18]. At first order of approximation, SDT distinguishes between *extrinsic* and *intrinsic* motivation. Motivated by extrinsic motivation, one engages in an action for instrumental reasons, whereas an action performed out of intrinsic motivation is done for its own purpose. Extrinsic motivation is driven by values and goals, whereas intrinsic motivation is based on emotions, such as joy and curiosity.

At second order of approximation, SDT distinguishes between four different forms of extrinsic motivation. One may picture those four so-called *regulations* as a continuum (Fig 1), depending on their *degree of internalisation*. Internalisation refers to the assimilation of a regulation with existing self-regulations, based on one's values and interests. In other words, the higher the degree of internalisation of a regulation, the more does one identify with the value or meaning of the resulting activity. *External regulation*, the type of extrinsic motivation with the least degree of internalization, refers to doing an activity to obtain rewards, avoid punishments and includes rule-following. Next on the spectrum, one finds *introjected regulation*, which refers to "the regulation of behaviour through self-worth contingencies such as ego-involvement and guilt" [17: p.2]. One acts out of introjected regulation to receive (self-)

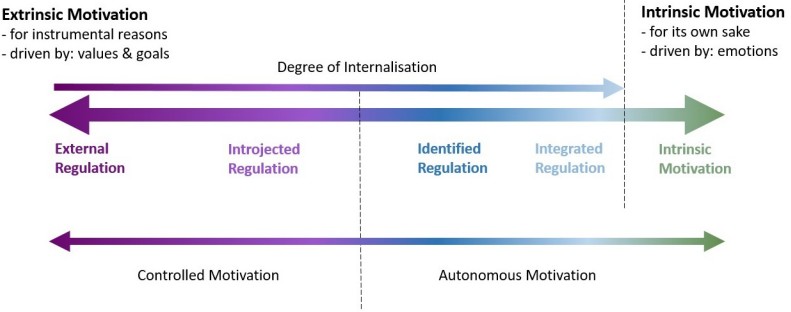

**Fig 1. The motivation continuum.** (Created by the author, based on [17,18]).

approval and avoid disapproval. Introjected regulation is based on assimilating a regulation in a way that it becomes internally pressuring, which means it is partially internalised, but remains *controlling*. Identified regulation is almost fully internalised and hence one identifies with the value or meaning of the engaged activity. The goal that the activity poses is accepted as one's own and as such has personal importance. Therefore, identified regulation is not controlled, but *autonomous*. The most autonomous and completely internalised form of extrinsic motivation is *integrated regulation*. It "refers to identifying with the value of an activity to the point that it becomes part of a person's habitual functioning and part of the person's sense of self." (ibid.: p.2).

It is important to note that in practice one engages in an activity out of a mix of these five forms of motivation (four forms of extrinsic motivation and intrinsic motivation), with one of them at peak. In other words, when one says an action is motivated by, for example, introjected regulation, then this is the type of motivation that was most determining making the decision to engage in the action. Research has found that each subscale of motivation correlates most positively with adjacent subscales and less positively (or more negatively) with non-adjacent ones [17].

The five different forms of motivation proposed by SDT make sense from a theoretical point of view, given the different degree of internalisation of the regulations and that intrinsic motivation is the only form of motivation not based on instrumental reasons. However, in practice, one may not always be able to (statistically) differentiate within autonomous forms of motivation (identified regulation, integrated regulation and intrinsic motivation) and controlled forms of regulation (external and introjected). Research has shown that there is a clear break in the consequences of controlled versus autonomous types of motivation, hence for some research questions these aggregates may be sufficient [17].

## 2.2. Reflexive motivation: Impact on behaviour

In analogy to [14], I relate autonomous motivational factors to those that stem from the internal component (values & personal motives) of the actor's situation and controlled motivational factors to those arising out of the three external components (material opportunities, norms & culture). These motivational factors influence the decisions made by the actor and hence their behaviour, according to RCT [13]. In other words, knowing which type of motivation is most present in a situation helps predicting the behavioural outcome. In turn, SDT-based organizational research (e.g. [18]) shows that acting out of different types of motivation yields different forms of impact on the actor. According to the authors [18], autonomous types of motivation are positively related to the satisfaction of psychological needs for autonomy. In SDT, "autonomy means endorsing one's actions at the highest level of reflection" [19]. This is one of three basic psychological needs for humans to function optimally, both in terms of their job-performance and their well-being. For example, autonomy is related to a higher job satisfaction, commitment, competence, personal initiative & involvement, effort, work motivation and less emotional distress, role stress, absenteeism and turnover intentions [17].

The prevailing organisational climate is an important factor to foster or inhibit autonomous types of motivation. For example, motivational job design has the potential to compensate for poor leadership and booster motivation levels [17]. In other words, if employees feel in control of the situation (through job autonomy), managerial constraints do not necessarily have negative effects on their motivation. Participative management, which naturally increases perceived job autonomy, is found to have a positive impact on internalisation of the importance of the task, and subsequently leads to a greater intrinsic motivation & performance and less strain (ibid.).

By contrast, controlled motivation, unsurprisingly, is shown to be unrelated to autonomy support and need satisfaction, but related to other types of controlling leadership behaviours [18]. While the authors also found that introjected regulation was positively related to aspired outcomes, when controlling for identified motivation, many of these relations disappeared. Generally, acting out of controlled motivation is less beneficial to individuals' optimal functioning than autonomous motivation and may even lead to unwanted behaviour, such as deviant or unethical actions (ibid.).

## 2.3. Research question & hypotheses

This study builds up on the qualitative study on structural conditions of research in astronomy [14], the quantitative study on the relationship between perceived publication pressure, organisational justice, overcommitment and research misbehaviour in astronomy [12], as well as previous studies on organisational psychology and SDT [17–19].

In the light of the previous research performed on the structural conditions of research in astronomy and the theoretical background as outlined above, I work from the assumption that the higher autonomous types of motivation the lower the likelihood to perceive publication pressure and the higher the likelihood to perceive organisational justice and to feel an overcommitment to work; and the other way round with controlled types of motivation. (Note: the relation between autonomous motivation and overcommitment is probably why the authors of [20] conflate overcommitment with "intrinsic drive".)

Therefore, I study the following three research questions and test the related hypotheses:

1. (RQ1): Is the motivation for astronomers to perform research congruent with their motivation to publish papers?

    a. (H1): Astronomers take on that profession more out of autonomous than controlled motivation.

    b. (H2): Astronomers' controlled motivation to publish is bigger than their autonomous motivation.

    c. (H3): Those for whom publication demands pose a more serious threat to their academic career (e.g., early-career & female researchers in a male-dominated field) will feel less autonomous motivation and more controlled motivation to publish.

2. (RQ2): Is there a difference in the effect of autonomous versus controlled types of motivation on turnover intentions and how much one loves their job?

    a. (H4): Higher autonomous motivation to become an astronomer and to publish decrease, while controlled motivation to become and astronomer and to publish increase the likelihood for astronomers to consider leaving academia/ quitting their job.

    b. (H5): Higher autonomous motivation to become an astronomer and to publish increase, while controlled motivation to become and astronomer and to publish decrease the likelihood for astronomers to report that they love their job.

3. (RQ3): Is there a difference in the effect of autonomous versus controlled types of motivation on perceived publication pressure, distributive justice, overcommitment and frequency of observed misbehaviour?

    a. (H6): The perception of publication pressure increases with increasing controlled motivation and decreases with increasing autonomous motivation to publish.

b. (H7): The perception of effort put into work increases with increasing controlled motivation and decreases with increasing autonomous motivation to publish.

c. (H8): The perception of reward obtained from one's work increases with decreasing controlled motivation and with increasing autonomous motivation to publish.

d. (H9): The perception of overcommitment increases with increasing autonomous motivation to become an astronomer and with an increasing controlled motivation, external, introjected and identified regulation to publish.

e. (H10): The perception of frequency of misconduct decreases with an autonomous motivation to become an astronomer and to publish and increases with a controlled motivation to become an astronomer, and a controlled motivation, external and introjected regulation to publish.

## 3. Materials and methods

### 3.1. Sample selection & procedure

This study is based on a web-based quantitative survey. The ideal aim was to run a census. However, because there is no official, complete list of all astronomers worldwide I used a multi-stage cluster sampling technique, targeting as many astronomers as possible. This consisted of distributing the survey invitation among astronomers from 176 universities, 56 non-academic research facilities & observatories and 17 societies & associations. In a consecutive stage, I asked the division heads of the International Astronomical Union (IAU) to distribute the survey invitation among their members. Five of nine division heads were non-responsive and so I reached the members of the remaining divisions through an automated script, based on publicly available email addresses. Based on the number of members of the IAU and the other institutions I contacted, I estimate that around 13,000–15,000 astronomers were reached in total. 3509 astronomers completed the survey at least partly, amounting to a response rate of roughly 25%, and 2011 astronomers completed the survey in full. A full description of the sample selection and procedure is described elsewhere [12].

The participants were informed about the content and the anonymity and voluntariness of participating in this study on the welcome page of the web-based survey. They were also informed that by clicking the button "Next" they would agree with the data protection regulation (https://www.hu-berlin.de/en/hu-en/imprint/data-protection-statement?set_language=en, accessed on 26th November 2021). An Institutional Review Board Statement is not applicable for this study.

The data relevant for this study is presented in S1 File. It includes the SPSS file (Heuritsch_-Survey2021_DATA.sav) of all cases and relevant variables, the Excel file (Heuritsch_Survey2021_Questions&Items) describing all questions, items and codes (with the ones not relevant to this study crossed out); and the Flowchart (Heuritsch_Survey2021_Flowchart.jpg) showing the possibilities of answer paths based on different filter questions.

### 3.2. Instruments

As outlined above, the survey is embedded in the conceptual framework of rational choice theory (RCT), organisational culture theory (OCT) and self-determination theory (SDT). This analysis is based on some instruments (and resulting data) used in [12], and I enhance their model (Fig 1 therein) with motivational factors as independent variables. Fig 2 shows a comparison of the initial model in [12] and the model used in this study. All variables used in this figure are described in the following sections 3.2.1 to 3.2.3.

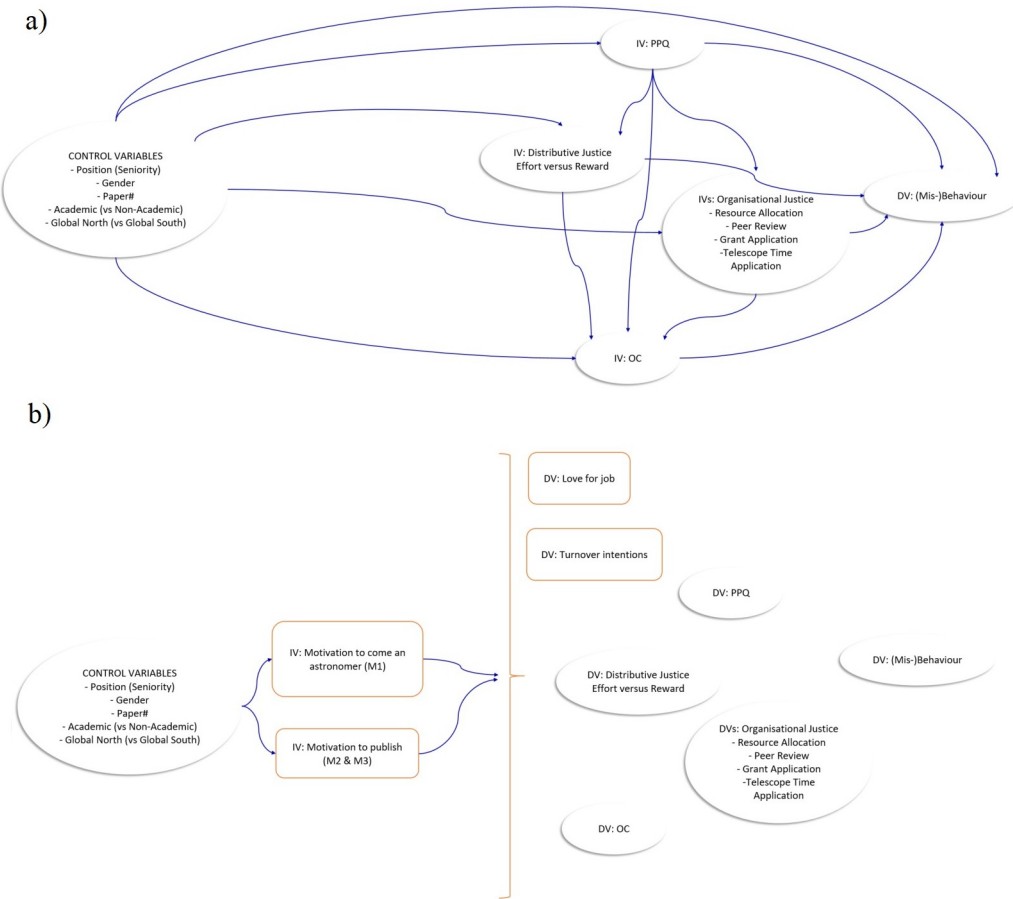

**Fig 2. Comparison of the model used in [12] and this study's enhanced model, representing control, independent (IV) and dependent variables (DV).** Arrows indicate that a construct (arrowhead) is being regressed onto another construct (arrow start). a) corresponds to Fig 1 in [12]; used with author's permission, and format slightly adapted for better comparability]. b) corresponds to the model used in this study. For better readability, the curly bracket substitutes the arrows going from each control variable and IV to each DV. Those IVs and DVs which enhance the model represented by a) are highlighted as a square (and orange) compared to the ovals used in a).

This study is based on the following instruments to measure the independent and dependent variables:

### 3.2.1 Dependent variables.

- Scientific (Mis-) Behaviour
  The instrument is described in detail elsewhere [12: p.7] and is originally based on [6,8,9,20,21]. It measures the perceived frequency of observed (mis-) behaviour and consists of 18 items.

- Turnover intentions & Love for job
  In addition to instruments used in the analysis performed in [12], in this analysis I use three more dependent variables: Turnover intentions of academics & of non-academics and astronomer's love for their job. To measure these, I asked to what extent respondents agree/disagree to the following statements: "I regularly contemplate leaving academia" (for those primarily employed in academia), "I regularly contemplate quitting my job" (for those primarily employed in non-academic institutions) and "I love my job". The scale ranged from "Strongly Disagree" (1) to "Strongly Agree" (5).

- Perceived Publication Pressure
  The PPQ; the instrument measuring perceived publication pressure is described in detail
  elsewhere [12: p.7] and is originally based on [22]. The cleaned version of the construct I
  used here consists of 12 items.

- Perceived Organisational Justice: Distributive & Procedural Justice
  We measure perceived *distributive justice* via the Effort-Reward Imbalance (ERI; [23])
  instrument. A detailed description of the instrument can be found elsewhere [12: p.7-8] and
  it consists of three effort and eight reward items. As for *procedural justice* (PJ), in this study I
  consider the following processes: a) Resource allocation, b) Peer review, c) Grant application
  and d) Telescope time application. The adaptation from the original instrument [20] can be
  found elsewhere [12: p.7-8].

- Perceived Overcommitment
  The instrument measuring perceived overcommitment (OC)–the inclination to overwork–is
  originally based on the ERI instrument [23] and the adaptation I used in this study can be
  found elsewhere [12: p.8].

### 3.2.2 Independent variables.

- Motivation to become an Astronomer (M1):
  We designed 8 items for this instrument, which are inspired by [18]. Five of these items
  intend to represent autonomous motivation and three items draw on controlled motivation,
  a classification I subsequently tested in a factor analysis (*section 4.1.*). The full instrument
  can be found Table S1 in S1 Appendix.

- Motivation to publish:
  We used two instruments to measure this concept. In order to measure the "drivers to publish" (M2) I designed 9 items. Five of these items intend to represent autonomous motivation and four items draw on controlled motivation, a classification I subsequently tested in
  a factor analysis (*section 4.1.*). The second instrument to measure the motivation publish
  does so by accounting for the feelings one experiences when not publishing the amount of
  papers that one aimed to publish (M3). This is because these emotions resulting from an
  experienced fail of creating publications give an indication for the motivation *why* one
  would want to publish. This instrument consists of 10 items in total, where four items each
  intend to measure introjected regulation and external regulation. One item draws on identified regulation and one item was used as a control item. Both instruments can be found
  Table S2a and S2b in S1 Appendix and are inspired by [18]. Note at this point that I did
  not include a subscale for integrated regulation in this study. This is because Gagné et al.
  [18] explain that it can hardly be statistically separated from identified and intrinsic motivation subscales.

### 3.2.3 Control variables.

Because I enhance Heuritsch's [12] model, I used the same role-associated and individual aspects as control variables: gender, academic position, whether
one is primarily employed at an academic versus non-academic institution, whether one is
employed at an institution in the global North/ South (https://meta.wikimedia.org/wiki/List_
of_countries_by_regional_classification, accessed on 26th November 2021), and number of
published first or co-author papers in the last 5 years. The reference categories are: gender:
female/ non-binary; academic position: full professor; primary employer: Non-Academic;
institute location: global South; Number of papers published in the last five years: 1–5.

**3.2.4 Additional instruments.** In addition to the dependent, independent and control variables, this study looks at two more aspects related to publication pressure and motivational aspects regarding work. I developed a nine-item battery to ask for the source of the perceived publication pressure Table S3 in S1 Appendix and asked respondents to rank the three most rewarding aspects of their work, given eight items Table S4 in S1 Appendix.

## 3.3. Statistical analyses

The analysis of the survey data for this study was performed in SPSS and R. A full description of the data preparation can be found elsewhere [12]. All instruments, measuring the independent variables, are scored on a scale from 1 (strongly disagree) to 5 (strongly agree) and are treated as continuous variables in this study's regression models. The steps to arrive at the final model started with testing the independent variable constructs PPQ, ERI (including overcommitment) and M1-M3 by performing an exploratory factor analysis (EFA) for ordinal data (CATPCA in SPSS) and testing the constructs' scale reliabilities based on Cronbach Alphas. For the EFA I used Promax Kaiser-normalisation for rotating the factors. Next, all independent variable constructs were tested by means of confirmatory factor analysis (CFA) using Lavaan version 0.5–23 (Rosseel, 2012) in R version 3.3.1. For each construct, I used the residual correlation matrices to determine significant correlations of the indicators and included them into the respective models. After checking for construct validity, I used a cleaned version of the PPQ, the effort-, reward- and overcommitment subscales and the factors of the motivational constructs as independent variables. Finally, I used SPSS linear regression analysis with listwise exclusion to test my hypotheses. The results section will present the results of the EFAs, CFAs, descriptive statistics, effects of the control variables on the motivational constructs and the regression models of turnover intentions & love for job, PPQ, ERI, OC and perceived misconduct being regressed onto the motivational factors and control variables.

## 4. Results

### 4.1 Exploratory & confirmatory factor analyses

The results of the EFAs & CFAs for the independent variables PPQ and ERI can be found elsewhere [12]. This section describes the results of the EFAs & CFAs for the variables related to motivation: "Motivation to become an Astronomer" (M1), "Drivers to publish" (M2) and "Feelings when failing to publish" (M3).

As for M1, I set the CATPCA to yield four factors to account for the four different types of motivation (external, introjected & identified regulation and intrinsic motivation). The results can be found in Table S1a in S2 Appendix and compared with the categorisation as expected from theory (see Table S1b in S2 Appendix). The theoretical categorisation turns out to be matching the outcome of the CATPCA apart from the fact that curiosity and needing a job were put together in one category. I henceforth denoted this the "residual category", which turned out to be a good choice, since the Cronbach alpha test (see Table S1c in S2 Appendix) for the whole construct showed that removing these two items would increase the reliability of the M1 construct. This resulted in 6 remaining items. The Cronbach alpha for the remaining M1 construct is 0.666 (see Table S1d in S2 Appendix), which is not optimal, but workable. I henceforth used this cleaned M1 for any further analysis. I performed another CATPCA yielding two factors (see Table S1e in S2 Appendix), which can be interpreted as M1F1: Autonomous Motivation to become an astronomer (comprising of identified regulation & intrinsic motivation) & M1F2: Controlled Motivation to become an astronomer (comprising of introjected regulation).

For M2 I also performed an exploratory 4-factor CATPCA (see Table S2a in S2 Appendix) in order to compare my categorisation of the types of motivation based on theory with the results from the EFA. Two items displayed a difference: an item theoretically categorised as introjected regulation was associated with external regulation by the CATPCA and an item theoretically classified as identified regulation was associated with intrinsic motivation (see Table S2b in S2 Appendix). This is not an issue however, since in both cases the categorisation within the higher abstraction level "Autonomous motivation" versus "Controlled motivation", which is used for subsequent analyses, remains valid as next step shows. The Cronbach Alpha for the M2 construct yields 0.662 (see Table S2c in S2 Appendix), which in analogy to the M1 construct is workable. A 2-factor CATPCA (see Table S2d in S2 Appendix) delivers M2F1: Autonomous Motivation to publish (comprising of identified regulation & intrinsic motivation) & M2F2: Controlled Motivation to publish (comprising of external & introjected regulation).

For M3 I performed an exploratory 3-factor CATPCA (see Table S3a in S2 Appendix) in order to compare my categorisation of the types of motivation based on theory with the results from the EFA. The comparison (see Table S3b in S2 Appendix) yields no difference. After removing the control item from the list, I performed an analysis of Cronbach's Alpha of the cleaned M3 construct, which amounts to 0.867 (see Table S3c in S2 Appendix). M3 therefore shows good internal consistency. Performing another 3-factor CATPCA (see Table S3d in S2 Appendix) for the cleaned M3 construct yields the following factors: M3F1: Identified Regulation to publish, M3F2: Introjected Regulation to publish & M3F3: External Regulation to publish.

We subsequently ran CFAs for all three of the motivation constructs M1-M3 and their factors, which results are presented in Table S4 in S2 Appendix. This table includes the model fit indices CFI and TLI, where >0.9 indicates a good fit for both and RMSEA, where <0.05 denotes a good fit. All independent variable constructs show a good fit according to CFI and TLI. As for RMSEA the fit is good for M1 and acceptable for the other two constructs. Pearson correlation coefficients show weak to moderate correlations between the various motivational factors, but strong correlations between M2F2 and M3F2 (0.413), between M2F2 and M3F3 (0.5) and between M3F2 and M3F3 (0.503). This makes sense, since the controlled motivation to publish (M2F2) should be congruent with the introjected (M3F2) and external (M3F3) to publish. Furthermore, M3F2 and M3F3 are adjacent subscales of the motivation continuum and are therefore also expected to correlate positively (cf. [17]).

## 4.2. Descriptive statistics

The descriptive statistics of the control variables (gender, academic position, primary employment at an academic/ non-academic institution, location of employment and number of published papers), independent variables (perceived publication pressure, perceived organisational distributive & procedural justice and perceived overcommitment), as well as the dependent variable perceived occurrence of misconduct can be found elsewhere [12].

Table 1 presents the mean scores of the motivational factors, resulting from the factor analyses of the independent variable motivation constructs "Motivation to become an Astronomer" (M1), "Drivers to publish" (M2) and "Feelings when failing to publish" (M3). In addition to these independent variables, Table 1 contains the mean scores of the dependent variables love for job and turnover intentions.

Table 1 shows that the autonomous motivation to become an astronomer is higher than the controlled one (means = 4.29 versus 3.05, respectively) and the opposite is true for the motivation to publish (means = 3.5 versus 3.76, respectively). These results support H1 & H2, respectively. H2 is additionally supported by the fact that the two forms of controlled motivation,

**Table 1. Means and standard deviations of the independent variables—The factors of the motivation constructs "Motivation to become an Astronomer" (M1), "Drivers to Publish" (M2) & "Feelings when failing to Publish" (M3)—And dependent variables—Turnover intentions & love for job.**

|  | n valid | Mean | SD |
|---|---|---|---|
| **Independent Variables** |  |  |  |
| M1F1: Autonomous Motivation to become an astronomer | 2503 | 4.29 | 0.63 |
| M1F2: Controlled Motivation to become an astronomer | 2496 | 3.05 | 1.16 |
| M2F1: Autonomous Motivation to publish | 2032 | 3.50 | 0.70 |
| M2F2: Controlled Motivation to publish | 2032 | 3.76 | 0.77 |
| M3F1: Identified Motivation to publish | 1902 | 2.81 | 1.18 |
| M3F2: Introjected Motivation to publish | 1856 | 3.64 | 1.03 |
| M3F3: External Motivation to publish | 1919 | 3.19 | 1.17 |
| **Dependent Variables** |  |  |  |
| Turnover Intentions (Academics) | 1467 | 2,53 | 1,45 |
| Turnover Intentions (Non-Academics) | 286 | 2,08 | 1,20 |
| Love for Job | 1759 | 4,13 | 0,92 |

N = 3509 survey respondents and "n valid" excludes missing data.

introjected & external regulation to publish are bigger drivers than identified regulation (means = 3.64, 3.19 & 2.81, respectively). For the sake of brevity, the detailed analysis regarding the dominating motivational factors to become an astronomer and to publish can be found in Table S1 –Table S3 in S3 Appendix. The same holds for the responses to the additional two instruments; the ranking question "What do you find most rewarding about your work?" and the multiple choice question regarding the source of perceived publication pressure (see Table S4 & S1 Fig in S3 Appendix, respectively).

## 4.3. Motivations & control variables

In this section I test H3 by regressing the motivational factors to publish (M2F1, M2F2, M3F1, M3F2 & M3F3) as dependent variables onto the control variables as independent ones (cf. Fig 2). The results regarding the regression models for the autonomous (M1F1) & controlled (M1F2) motivation to become an astronomer in dependence of the control variables can be found in Table S5 in S3 Appendix.

The autonomous motivation to publish (Table 2; M2F1) tends to be lower for males as compared to females/ non-binaries (by 0.095 points). Positions ranked lower than Full Professor and "Other" are also likely to feel less autonomous motivation to publish as compared to Full Professors, except from Associate Professors, whose effect is not statistically significant. Astronomers who have published more than 20 papers in the last 5 years feel more autonomous motivation to publish than those who have published 1 to 5 papers. Those employed by an institution in the Global North report less autonomous motivation to publish than those on the Global South (by 0.328 points).

The controlled motivation to publish (M2F2) also tends to be lower for males than females/ non-binaries (by 0.225 points). Contrary to the autonomous motivation to publish, positions ranked lower than Full Professor tend to perceive more controlled motivation to publish than Full Professors. Astronomers who haven't published any papers in the last 5 years feel less controlled motivation to publish than those having published 1 to 5 papers (by 0.257 points).

Identified regulation to publish (Table 3; M3F1) tends to be lower for males than females/ non-binaries (by 0.217 points) as well as for PhD Candidates as compared to Full Professors (by 0.307 points).

**Table 2. Regression models of DV = M2F1 & M2F2 (both N = 1361); regressed onto the control variables.**

| | M2F1: Autonomous Motivation to publish | | | | | M2F2: Controlled Motivation to publish | | | | |
| --- | --- | --- | --- | --- | --- | --- | --- | --- | --- | --- |
| | Unstandardized Coefficients | | Stand. Coeff. | t | Sig. | Unstandardized Coefficients | | Stand. Coeff. | t | Sig. |
| | B | Std. Error | Beta | | | B | Std. Error | Beta | | |
| Intercept: | 3.984 | 0.104 | | 38.269 | <0.001 * | 3.756 | 0.113 | | 33.261 | <0.001 * |
| Gender: Male | -0.095 | 0.041 | -0.062 | -2.309 | 0.021 * | -0.225 | 0.045 | -0.135 | -5.034 | <0.001 * |
| Position: PhD Candidate | -0.268 | 0.074 | -0.132 | -3.640 | <0.001 * | 0.405 | 0.080 | 0.183 | 5.077 | <0.001 * |
| Position: Postdoc | -0.259 | 0.054 | -0.158 | -4.830 | <0.001 * | 0.386 | 0.058 | 0.216 | 6.648 | <0.001 * |
| Position: Assistant Prof. | -0.160 | 0.072 | -0.064 | -2.209 | 0.027 * | 0.261 | 0.078 | 0.096 | 3.331 | 0.001 * |
| Position: Associate Prof. | -0.109 | 0.060 | -0.055 | -1.827 | 0.068 | 0.147 | 0.065 | 0.068 | 2.282 | 0.023 * |
| Position: Other | -0.199 | 0.063 | -0.098 | -3.149 | 0.002 * | -0.071 | 0.068 | -0.032 | -1.043 | 0.297 |
| Primary Employer: Academic | -0.046 | 0.079 | -0.016 | -0.586 | 0.558 | 0.076 | 0.085 | 0.024 | 0.895 | 0.371 |
| Papers published: Submission | -0.092 | 0.130 | -0.020 | -0.705 | 0.481 | -0.046 | 0.141 | -0.009 | -0.328 | 0.743 |
| Papers published: 0 | 0.090 | 0.105 | 0.025 | 0.862 | 0.389 | -0.257 | 0.113 | -0.064 | -2.268 | 0.023 * |
| Papers published: 6–20 | 0.063 | 0.049 | 0.042 | 1.296 | 0.195 | 0.037 | 0.053 | 0.023 | 0.703 | 0.482 |
| Papers published: >20 | 0.159 | 0.051 | 0.108 | 3.149 | 0.002 * | 0.038 | 0.055 | 0.023 | 0.685 | 0.494 |
| Location: Global North | -0.328 | 0.050 | -0.176 | -6.589 | <0.001 * | -0.087 | 0.054 | -0.042 | -1.602 | 0.109 |

* indicates statistical significance (p < 0.05).

**Table 3. Regression models of DV = M3F1 (N = 1305), M3F2 (N = 1294) & M3F3 (N = 1265); regressed onto the control variables.**

| | M3F1: Identified Regulation to publish | | | | | M3F2: Introjected Regulation to publish | | | | | M3F3: External Regulation to publish | | | | |
| --- | --- | --- | --- | --- | --- | --- | --- | --- | --- | --- | --- | --- | --- | --- | --- |
| | Unstandardized Coefficients | | Stand. Coeff. | t | Sig. | Unstandardized Coefficients | | Stand. Coeff. | t | Sig. | Unstandardized Coefficients | | Stand. Coeff. | t | Sig. |
| | B | Std. Error | Beta | | | B | Std. Error | Beta | | | B | Std. Error | Beta | | |
| Intercept: | 3.515 | 0.182 | | 19.268 | <0.001 * | 2.713 | 0.178 | | 15.276 | <0.001 * | 3.277 | 0.152 | | 21.618 | <0.001 * |
| Gender: Male | -0.217 | 0.072 | -0.085 | -2.996 | 0.003 * | -0.416 | 0.070 | -0.161 | -5.950 | <0.001 * | -0.399 | 0.060 | -0.176 | -6.679 | <0.001 * |
| Position: PhD Candidate | -0.307 | 0.130 | -0.090 | -2.368 | 0.018 * | 0.450 | 0.125 | 0.131 | 3.609 | <0.001 * | 0.506 | 0.108 | 0.164 | 4.677 | <0.001 * |
| Position: Postdoc | -0.100 | 0.094 | -0.037 | -1.059 | 0.290 | 0.496 | 0.091 | 0.181 | 5.461 | <0.001 * | 0.774 | 0.077 | 0.323 | 10.046 | <0.001 * |
| Position: Assistant Prof. | -0.013 | 0.127 | -0.003 | -0.099 | 0.921 | 0.390 | 0.122 | 0.094 | 3.199 | 0.001 * | 0.772 | 0.104 | 0.211 | 7.447 | <0.001 * |
| Position: Associate Prof. | -0.048 | 0.105 | -0.015 | -0.454 | 0.650 | 0.170 | 0.102 | 0.051 | 1.672 | 0.095 | 0.404 | 0.086 | 0.140 | 4.705 | <0.001 * |
| Position: Other | -0.182 | 0.111 | -0.054 | -1.640 | 0.101 | 0.162 | 0.108 | 0.047 | 1.492 | 0.136 | 0.165 | 0.093 | 0.054 | 1.773 | 0.077 |
| Primary Employer: Academic | -0.135 | 0.138 | -0.028 | -0.983 | 0.326 | 0.244 | 0.134 | 0.050 | 1.829 | 0.068 | 0.163 | 0.115 | 0.038 | 1.412 | 0.158 |
| Papers published: Submission | 0.143 | 0.228 | 0.019 | 0.625 | 0.532 | 0.538 | 0.219 | 0.071 | 2.453 | 0.014 * | 0.466 | 0.195 | 0.068 | 2.386 | 0.017 * |
| Papers published: 0 | 0.301 | 0.188 | 0.047 | 1.600 | 0.110 | -0.043 | 0.179 | -0.007 | -0.242 | 0.809 | -0.051 | 0.171 | -0.008 | -0.300 | 0.764 |
| Papers published: 6–20 | 0.014 | 0.085 | 0.006 | 0.170 | 0.865 | -0.105 | 0.082 | -0.042 | -1.281 | 0.200 | -0.072 | 0.070 | -0.033 | -1.027 | 0.305 |
| Papers published: >20 | -0.084 | 0.089 | -0.034 | -0.940 | 0.347 | -0.310 | 0.086 | -0.123 | -3.602 | <0.001 * | -0.176 | 0.073 | -0.080 | -2.413 | 0.016 * |
| Location: Global North | 0.094 | 0.087 | 0.030 | 1.077 | 0.282 | 0.041 | 0.085 | 0.013 | 0.484 | 0.629 | 0.213 | 0.072 | 0.077 | 2.967 | 0.003 * |

* indicates statistical significance (p < 0.05).

Introjected regulation to publish (M3F2) is likely to be lower for males than females/non-binaries (by 0.416 points). PhD Candidates, Postdocs and Assistant Professors tend to have a higher introjected regulation to publish than Full Professors. Astronomers who currently have their first paper in the submission process feel a higher introjected motivation to publish than those who have published 1–5 papers in the last five years (by 0.538 points) and those who have published more than 20 papers feel a lower introjected motivation (by 0.310 points).

External regulation to publish (M3F3) also tends to be lower for males than females/non-binaries (by 0.399 points) and higher for all positions ranked lower than Full Professors. As with introjected motivation to publish, astronomers whose first paper is currently in the submission process feel a higher external regulation to publish (by 0.466 points) and those who have published more than 20 papers in the last five years feel a lower external regulation than the reference category of 1 to 5 papers (by 0.176 points). Astronomers being employed in the Global North tend to experience a higher external motivation to publish (by 0.213 points) than those in the Global South.

## 4.4. Motivations & turnover intentions and love for job

In this section I test H4 & H5. I study how motivational factors to become an astronomer and to publish affect the likelihood of astronomers feeling like quitting (H4) or loving (H5) their job.

The regression analysis on which of the seven motivational factors play a role in regularly contemplating leaving academia yield the following results (Table 4): Those astronomers who

**Table 4. Regression model of DV = Turnover intentions of astronomers who are primarily employed at an academic institution (N = 1147) versus at a non-academic (N = 224) institution; regressed onto the seven motivational factors and the control variables.**

| | Turnover Intentions (Academics) | | | | | Turnover Intentions (Non-Academics) | | | | |
|---|---|---|---|---|---|---|---|---|---|---|
| | Unstandardized Coefficients | | Stand. Coeff. | t | Sig. | Unstandardized Coefficients | | Stand. Coeff. | t | Sig. |
| | B | Std. Error | Beta | | | B | Std. Error | Beta | | |
| Intercept: | 2.485 | 0.355 | | 6.990 | <0.001 * | 2.394 | 0.804 | | 2.976 | 0.003 * |
| M1F1: Aut. mot. astronomer | -0.283 | 0.061 | -0.121 | -4.666 | <0.001 * | -0.040 | 0.133 | -0.021 | -0.304 | 0.762 |
| M1F2: Contr. mot. astronomer | -0.026 | 0.033 | -0.020 | -0.789 | 0.430 | -0.023 | 0.075 | -0.021 | -0.307 | 0.760 |
| M2F1: Aut. mot. publishing | -0.239 | 0.058 | -0.113 | -4.101 | <0.001 * | -0.109 | 0.137 | -0.058 | -0.794 | 0.428 |
| M2F2: Contr. mot. publishing | 0.082 | 0.058 | 0.042 | 1.417 | 0.157 | 0.027 | 0.152 | 0.017 | 0.180 | 0.857 |
| M3F1: Identified reg. publishing | -0.040 | 0.033 | -0.032 | -1.231 | 0.218 | -0.012 | 0.079 | -0.012 | -0.153 | 0.879 |
| M3F2: Introjected reg. publishing | 0.176 | 0.037 | 0.144 | 4.804 | <0.001 * | 0.079 | 0.091 | 0.074 | 0.863 | 0.389 |
| M3F3: External reg. publishing | 0.291 | 0.045 | 0.207 | 6.472 | <0.001 * | 0.162 | 0.102 | 0.145 | 1.593 | 0.113 |
| Gender: Male | 0.068 | 0.083 | 0.021 | 0.829 | 0.408 | -0.237 | 0.177 | -0.092 | -1.337 | 0.183 |
| Position: PhD Candidate | 0.873 | 0.148 | 0.202 | 5.912 | <0.001 * | Variable non-existent for non-academic astronomers | | | | |
| Position: Postdoc | 0.996 | 0.107 | 0.300 | 9.269 | <0.001 * | Variable non-existent for non-academic astronomers | | | | |
| Position: Assistant Prof. | 0.294 | 0.140 | 0.058 | 2.100 | 0.036 * | Variable non-existent for non-academic astronomers | | | | |
| Position: Associate Prof. | 0.231 | 0.115 | 0.057 | 2.005 | 0.045 * | Variable non-existent for non-academic astronomers | | | | |
| Position: Other | 0.322 | 0.130 | 0.068 | 2.475 | 0.013 * | Variable non-existent for non-academic astronomers | | | | |
| Papers published: Submission | 0.480 | 0.258 | 0.050 | 1.860 | 0.063 | -0.671 | 0.841 | -0.053 | -0.798 | 0.426 |
| Papers published: 0 | -0.547 | 0.231 | -0.061 | -2.370 | 0.018 * | -0.790 | 0.437 | -0.122 | -1.805 | 0.072 |
| Papers published: 6–20 | -0.097 | 0.093 | -0.032 | -1.047 | 0.295 | -0.325 | 0.197 | -0.126 | -1.646 | 0.101 |
| Papers published: >20 | -0.197 | 0.099 | -0.063 | -1.985 | 0.047 * | -0.572 | 0.202 | -0.222 | -2.834 | 0.005 * |
| Location: Global North | 0.016 | 0.098 | 0.004 | 0.163 | 0.871 | -0.036 | 0.308 | -0.008 | -0.118 | 0.906 |

* indicates statistical significance (p < 0.05).

**Table 5. Regression model of DV = Love for Job; regressed onto the seven motivational factors and the control variables.**

| | | Unstandardized Coefficients | | Standardized Coefficients | t | Sig. |
|---|---|---|---|---|---|---|
| | | B | Std. Error | Beta | | |
| Love for Job | Intercept: | 2.690 | 0.262 | | 10.261 | <0.001 * |
| | M1F1: Aut. mot. astronomer | 0.268 | 0.041 | 0.178 | 6.520 | <0.001 * |
| | M1F2: Contr. mot. astronomer | 0.049 | 0.023 | 0.059 | 2.169 | 0.03 * |
| | M2F1: Aut. mot. publishing | 0.259 | 0.040 | 0.189 | 6.482 | <0.001 * |
| | M2F2: Contr. mot. publishing | 0.001 | 0.040 | 0.001 | 0.030 | 0.976 |
| | M3F1: Identified reg. publishing | 0.059 | 0.023 | 0.073 | 2.614 | 0.009 * |
| | M3F2: Introjected reg. publishing | -0.139 | 0.025 | -0.174 | -5.487 | <0.001 * |
| | M3F3: External reg. publishing | -0.086 | 0.031 | -0.094 | -2.786 | 0.005 * |
| | Gender: Male | 0.004 | 0.057 | 0.002 | 0.063 | 0.949 |
| | Position: PhD Candidate | -0.132 | 0.101 | -0.047 | -1.312 | 0.190 |
| | Position: Postdoc | 0.023 | 0.074 | 0.011 | 0.308 | 0.758 |
| | Position: Assistant Prof. | 0.102 | 0.097 | 0.031 | 1.047 | 0.295 |
| | Position: Associate Prof. | 0.019 | 0.080 | 0.007 | 0.243 | 0.808 |
| | Position: Other | 0.066 | 0.086 | 0.024 | 0.772 | 0.440 |
| | Primary Employer: Academic | -0.186 | 0.105 | -0.047 | -1.768 | 0.077 |
| | Papers published: Submission | -0.493 | 0.176 | -0.080 | -2.792 | 0.005 * |
| | Papers published: 0 | 0.269 | 0.156 | 0.047 | 1.728 | 0.084 |
| | Papers published: 6–20 | 0.013 | 0.064 | 0.006 | 0.198 | 0.843 |
| | Papers published: >20 | -0.001 | 0.068 | -0.001 | -0.017 | 0.986 |
| | Location: Global North | -0.120 | 0.068 | -0.048 | -1.782 | 0.075 |

* indicates statistical significance (p < 0.05).

feel more autonomous motivation to become an astronomer and more autonomous motivation to publish feel significantly less often that they want to leave academia (by 0.283 & 0.239 points, respectively). In contrast, those who feel more introjected and external regulation to publish are more likely to consider moving on from academia (by 0.176 & 0.291 points, respectively). Perhaps unsurprisingly, astronomers from other positions than Full Professor have a much higher chance of leaving academia. Astronomers who haven't published a paper in the last five years, or those who have published more than 20 papers, tend to have less turnover intentions than those of the reference category (1–5 papers).

The regression analysis on which of the seven motivational factors play a role in regularly contemplating quitting their job at their non-academic primary employer yield no statistically significant results for the motivational factors. However, I find that astronomers who have published more than 20 papers in the last five years are less likely (by 0.572 points) to think about a turnover as compared to those of the reference category (1–5 papers).

The regression analysis on which of the seven motivational factors play a role in loving their job (whether academic or non-academic) yield the following results (Table 5; N = 1226): Those who feel more autonomous motivation to become an astronomer and to publish are more likely to love their job (by 0.268 & 0.259 points, respectively). The same goes for those who report more controlled motivation to become an astronomer and more identified regulation to publish (by 0.049 & 0.059 points, respectively). The opposite is true for astronomers who have a higher introjected or external regulation to publish (by 0.139 & 0.086 points, respectively)–these tend to love their jobs less. An astronomer whose paper is currently in the submission process also tends to love their job less (by 0.493 points).

### 4.5. Motivations & PPQ, ERI, OC

We enhanced the regression models of the analysis performed in [12] with the seven motivational factors as independent variables (cf. Fig 2) and thereby test hypotheses H6 to H9 in this section.

The perception of publication pressure (Table 6; N = 1212) significantly increases with increasing autonomous motivation to become an astronomer (by 0.098 points) and introjected & external regulation to publish (by 0.116 & 0.292 points, respectively). By contrast, astronomers who feel more autonomous motivation and identified regulation to publish tend to perceive less publication pressure (by 0.107 & 0.049 points, respectively).

As for the distributive justice factors reward and effort (Table 7; N = 1207 & N = 1208, respectively), the perception of obtaining rewards from one's job increases with a higher controlled motivation to become an astronomer (by 0.041 points) and a higher autonomous & controlled motivation to publish (by 0.071 & 0.070 points, respectively). By contrast, respondents perceive their work less rewarding when they show a higher introjected and external regulation to publish (by 0.062 & 0.103 points, respectively). The perception of effort put into one's job significantly increases when respondents feel a higher controlled motivation, introjected & external regulation to publish (by 0.104, 0.089 & 0.132 points, respectively). By contrast, one tends to feel less need to put effort when one perceives a higher autonomous motivation to publish (by 0.135 points). The trend observed in the model in [12] that Full Professors tend to feel better rewarded than an astronomer from any other position is also observed here, although with larger effects. At the same time, ECRs tend to feel putting less effort into their work. Interestingly, astronomers, who have not published any papers in the

**Table 6. Regression model of DV = Perceived Publication Pressure; regressed onto the seven motivational factors and the control variables.**

| | | Unstandardized Coefficients | | Standardized Coefficients | t | Sig. |
|---|---|---|---|---|---|---|
| | | B | Std. Error | Beta | | |
| **Perceived Publication Pressure (PPQ)** | Intercept: | 2.266 | 0.199 | | 11.396 | <0.001 * |
| | M1F1: Aut. mot. astronomer | 0.098 | 0.031 | 0.080 | 3.140 | 0.002 * |
| | M1F2: Contr. mot. astronomer | -0.009 | 0.017 | -0.013 | -0.526 | 0.599 |
| | M2F1: Aut. mot. publishing | -0.107 | 0.030 | -0.096 | -3.509 | <0.001 * |
| | M2F2: Contr. mot. publishing | -0.028 | 0.030 | -0.027 | -0.933 | 0.351 |
| | M3F1: Identified reg. publishing | -0.049 | 0.017 | -0.075 | -2.887 | 0.004 * |
| | M3F2: Introjected reg. publishing | 0.116 | 0.019 | 0.177 | 6.034 | <0.001 * |
| | M3F3: External reg. publishing | 0.292 | 0.023 | 0.391 | 12.501 | <0.001 * |
| | Gender: Male | -0.029 | 0.043 | -0.017 | -0.685 | 0.494 |
| | Position: PhD Candidate | 0.108 | 0.077 | 0.046 | 1.396 | 0.163 |
| | Position: Postdoc | 0.084 | 0.056 | 0.047 | 1.497 | 0.135 |
| | Position: Assistant Prof. | 0.057 | 0.073 | 0.021 | 0.779 | 0.436 |
| | Position: Associate Prof. | -0.061 | 0.060 | -0.028 | -1.007 | 0.314 |
| | Position: Other | 0.029 | 0.065 | 0.013 | 0.447 | 0.655 |
| | Primary Employer: Academic | 0.142 | 0.079 | 0.045 | 1.800 | 0.072 |
| | Papers published: Submission | -0.067 | 0.135 | -0.013 | -0.495 | 0.621 |
| | Papers published: 0 | -0.054 | 0.129 | -0.010 | -0.420 | 0.675 |
| | Papers published: 6–20 | -0.053 | 0.048 | -0.033 | -1.099 | 0.272 |
| | Papers published: >20 | -0.134 | 0.051 | -0.082 | -2.617 | 0.009 * |
| | Location: Global North | -0.373 | 0.052 | -0.180 | -7.214 | <0.001 * |

* indicates statistical significance (p < 0.05).

**Table 7. Regression models of DV = Distributive Justice in terms of Reward & Effort; regressed onto the seven motivational factors, perceived publication pressure, and the control variables.**

| | Distributive Justice (DJ): Reward | | | | | Distributive Justice (DJ): Effort | | | | |
| --- | --- | --- | --- | --- | --- | --- | --- | --- | --- | --- |
| | Unstandardized Coefficients | | Stand. Coeff. | t | Sig. | Unstandardized Coefficients | | Stand. Coeff. | t | Sig. |
| | B | Std. Error | Beta | | | B | Std. Error | Beta | | |
| Intercept: | 4.256 | 0.239 | | 17.840 | <0.001 * | 1.914 | 0.234 | | 8.182 | <0.001 * |
| M1F1: Aut. mot. astronomer | 0.009 | 0.036 | 0.006 | 0.241 | 0.810 | 0.059 | 0.035 | 0.045 | 1.676 | 0.094 |
| M1F2: Contr. mot. astronomer | 0.041 | 0.019 | 0.056 | 2.126 | 0.034 * | -0.034 | 0.019 | -0.047 | -1.753 | 0.080 |
| M2F1: Aut. mot. publishing | 0.071 | 0.035 | 0.057 | 2.031 | 0.042 * | -0.135 | 0.034 | -0.113 | -3.930 | <0.001 * |
| M2F2: Contr. mot. publishing | 0.070 | 0.034 | 0.061 | 2.063 | 0.039 * | 0.104 | 0.033 | 0.095 | 3.106 | 0.002 * |
| M3F1: Identified reg. publishing | 0.020 | 0.020 | 0.027 | 1.018 | 0.309 | 0.032 | 0.019 | 0.045 | 1.658 | 0.098 |
| M3F2: Introjected reg. publishing | -0.062 | 0.022 | -0.085 | -2.791 | 0.005 * | 0.089 | 0.022 | 0.127 | 4.078 | <0.001 * |
| M3F3: External reg. publishing | -0.103 | 0.028 | -0.124 | -3.645 | <0.001 * | 0.132 | 0.028 | 0.165 | 4.726 | <0.001 * |
| PPQ: Perceived Pub. Pressure | -0.325 | 0.033 | -0.292 | -9.831 | <0.001 * | 0.293 | 0.032 | 0.275 | 9.047 | <0.001 * |
| Gender: Male | -0.044 | 0.049 | -0.024 | -0.909 | 0.363 | -0.016 | 0.048 | -0.009 | -0.329 | 0.743 |
| Position: PhD Candidate | -0.451 | 0.088 | -0.173 | -5.109 | <0.001 * | -0.394 | 0.087 | -0.157 | -4.543 | <0.001 * |
| Position: Postdoc | -0.456 | 0.064 | -0.233 | -7.135 | <0.001 * | -0.387 | 0.063 | -0.205 | -6.165 | <0.001 * |
| Position: Assistant Prof. | -0.271 | 0.084 | -0.090 | -3.242 | 0.001 * | -0.033 | 0.082 | -0.012 | -0.407 | 0.684 |
| Position: Associate Prof. | -0.223 | 0.069 | -0.093 | -3.255 | 0.001 * | -0.030 | 0.067 | -0.013 | -0.445 | 0.657 |
| Position: Other | -0.176 | 0.074 | -0.070 | -2.391 | 0.017 * | -0.172 | 0.072 | -0.071 | -2.384 | 0.017 * |
| Primary Employer: Academic | -0.037 | 0.090 | -0.010 | -0.408 | 0.683 | -0.011 | 0.088 | -0.003 | -0.129 | 0.897 |
| Papers published: Submission | 0.059 | 0.154 | 0.010 | 0.381 | 0.703 | -0.379 | 0.151 | -0.070 | -2.515 | 0.012 * |
| Papers published: 0 | 0.301 | 0.147 | 0.052 | 2.047 | 0.041 * | 0.025 | 0.144 | 0.004 | 0.170 | 0.865 |
| Papers published: 6–20 | 0.034 | 0.055 | 0.019 | 0.610 | 0.542 | -0.061 | 0.054 | -0.035 | -1.124 | 0.261 |
| Papers published: >20 | 0.108 | 0.058 | 0.060 | 1.847 | 0.065 | 0.096 | 0.057 | 0.055 | 1.666 | 0.096 |
| Location: Global North | 0.040 | 0.060 | 0.017 | 0.662 | 0.508 | 0.048 | 0.059 | 0.022 | 0.807 | 0.420 |

* indicates statistical significance (p < 0.05).

last five years, feel obtaining more reward from and putting less effort into their jobs than those having published 1–5 papers in the same period.

Enhancing the regression model for perceived overcommitment with the seven motivational factors yields the results presented in Table 8 (Model 1, N = 591,): Respondents with a higher introjected and identified regulation to publish feel more overcommitment (by 0.110 & 0.078 points; respectively). Taking perceived publication pressure and the perceived distributive & organizational justice items out of the equation (Table 8; Model 2, N = 1226), I obtain significant effects for all motivational factors, apart from controlled motivation to become an astronomer and controlled motivation to publish. Perceived overcommitment then increases when respondents report a higher autonomous motivation to become an astronomer (by 0.116 points), and a higher introjected, external and identified regulation to publish (by 0.171, 0.2 and 0.049 points, respectively). Autonomous motivation to publish decreases the likelihood of perceived overcommitment (0.097 points).

## 4.6. Motivations & scientific misbehaviour

The final step involved testing H10 by regressing the seven motivational factors and the control variables on frequency of misbehavior occurrence (Table 9; Model 2, N = 1208). I observe that controlled motivation to become an astronomer and autonomous motivation to publish decrease the observation of misbehaviour (by 0.038 & 0.067 points; respectively). By contrast,

**Table 8. Regression models of DV = Overcommitment.** Model 1: DV is regressed onto the seven motivational factors, perceived publication pressure, distributive & organizational justice, and the control variables. Model 2: DV is regressed onto the seven motivational factors and the control variables.

| | Over-commitment (OC) Model 1 | | | | | Over-commitment (OC) Model 2 | | | | |
|---|---|---|---|---|---|---|---|---|---|---|
| | Unstandardized Coefficients | | Stand. Coeff. | t | Sig. | Unstandardized Coefficients | | Stand. Coeff. | t | Sig. |
| | B | Std. Error | Beta | | | B | Std. Error | Beta | | |
| Intercept: | 1.293 | 0.429 | | 3.016 | 0.003 * | 2.131 | 0.250 | | 8.522 | <0.001 * |
| M1F1: Aut. mot. astronomer | 0.057 | 0.051 | 0.040 | 1.113 | 0.266 | 0.116 | 0.039 | 0.082 | 2.946 | 0.003 * |
| M1F2: Contr. mot. astronomer | 0.004 | 0.028 | 0.005 | 0.145 | 0.885 | -0.034 | 0.022 | -0.043 | -1.558 | 0.120 |
| M2F1: Aut. mot. publishing | -0.068 | 0.053 | -0.050 | -1.284 | 0.200 | -0.097 | 0.038 | -0.076 | -2.554 | 0.011 * |
| M2F2: Contr. mot. publishing | -0.018 | 0.047 | -0.015 | -0.370 | 0.711 | 0.008 | 0.038 | 0.007 | 0.205 | 0.838 |
| M3F1: Identified reg. publishing | 0.078 | 0.027 | 0.105 | 2.873 | 0.004 * | 0.049 | 0.021 | 0.065 | 2.293 | 0.022 * |
| M3F2: Introjected reg. publishing | 0.110 | 0.030 | 0.152 | 3.671 | <0.001 * | 0.171 | 0.024 | 0.228 | 7.091 | <0.001 * |
| M3F3: External reg. publishing | 0.024 | 0.041 | 0.027 | 0.583 | 0.560 | 0.200 | 0.029 | 0.233 | 6.789 | <0.001 * |
| PPQ: Perceived Pub. Pressure | 0.199 | 0.050 | 0.177 | 3.961 | <0.001 * | *not included in Model 2* | | | | |
| DJ: Effort | 0.396 | 0.043 | 0.350 | 9.185 | <0.001 * | *not included in Model 2* | | | | |
| DJ: Reward | -0.059 | 0.047 | -0.059 | -1.251 | 0.211 | *not included in Model 2* | | | | |
| OJ: Resource Allocation | -0.031 | 0.045 | -0.030 | -0.681 | 0.496 | *not included in Model 2* | | | | |
| OJ: Peer Review | 0.042 | 0.047 | 0.036 | 0.903 | 0.367 | *not included in Model 2* | | | | |
| OJ: Grant Application | -0.012 | 0.044 | -0.011 | -0.261 | 0.794 | *not included in Model 2* | | | | |
| OJ: Telescope Application | -0.071 | 0.048 | -0.059 | -1.476 | 0.140 | *not included in Model 2* | | | | |
| Gender: Male | -0.020 | 0.068 | -0.011 | -0.294 | 0.769 | -0.021 | 0.054 | -0.011 | -0.388 | 0.698 |
| Position: PhD Candidate | 0.166 | 0.168 | 0.037 | 0.988 | 0.324 | -0.094 | 0.096 | -0.036 | -0.982 | 0.327 |
| Position: Postdoc | -0.034 | 0.091 | -0.017 | -0.377 | 0.707 | -0.154 | 0.071 | -0.076 | -2.184 | 0.029 * |
| Position: Assistant Prof. | -0.050 | 0.104 | -0.018 | -0.483 | 0.629 | -0.158 | 0.093 | -0.051 | -1.707 | 0.088 |
| Position: Associate Prof. | -0.195 | 0.088 | -0.086 | -2.223 | 0.027 * | -0.179 | 0.076 | -0.072 | -2.354 | 0.019 * |
| Position: Other | -0.109 | 0.100 | -0.043 | -1.087 | 0.278 | -0.183 | 0.082 | -0.069 | -2.238 | 0.025 * |
| Primary Employer: Academic | -0.054 | 0.115 | -0.016 | -0.467 | 0.641 | 0.072 | 0.100 | 0.020 | 0.718 | 0.473 |
| Papers published: Submission | *Excluded (filtered due to OJ: Peer Review)* | | | | | -0.324 | 0.168 | -0.056 | -1.926 | 0.054 |
| Papers published: 0 | *Excluded (filtered due to OJ: Peer Review)* | | | | | -0.188 | 0.149 | -0.035 | -1.266 | 0.206 |
| Papers published: 6–20 | 0.070 | 0.082 | 0.038 | 0.851 | 0.395 | 0.065 | 0.061 | 0.034 | 1.059 | 0.290 |
| Papers published: >20 | 0.082 | 0.081 | 0.048 | 1.019 | 0.309 | 0.177 | 0.065 | 0.094 | 2.745 | 0.006 * |
| Location: Global North | 0.024 | 0.088 | 0.010 | 0.272 | 0.785 | -0.148 | 0.064 | -0.062 | -2.294 | 0.022 * |

* indicates statistical significance ($p < 0.05$).

introjected & external regulation to publish increase the perceived frequency of misbehaviour occurrence (by 0.085 & 0.144 points; respectively). When adding the PPQ, ERI & PJ, OC constructs to the regression model (Table 9; Model 1, N = 586), as done in [12], the effects of the motivational factors yield no statistical significance anymore, apart from the controlled motivation to become an astronomer (decrease of 0.045 points), emphasising the mediating effect that the cultural constructs have (*section 4.5*.). In contrast to the analysis in [12], increased perceived procedural justice in terms of peer review also decreases the likelihood to observe misbehaviour (0.093 points).

## 5. Discussion

This study's quantitative results supports findings of the qualitative research performed in [14] and I find evidence supporting my hypotheses. The author (ibid.) found that astronomers'

**Table 9. Regression models of DV = perception of frequency of misbehaviour occurrence.** Model 1: DV is regressed onto the seven motivational factors, perceived publication pressure, distributive justice & organizational, overcommitment, and the control variables. Model 2: DV is regressed onto the seven motivational factors and the control variables.

| | Perception of misbehaviour occurrence Model 1 | | | | | Perception of misbehaviour occurrence Model 2 | | | | |
| | Unstandardized Coefficients | | Stand. Coeff. | t | Sig. | Unstandardized Coefficients | | Stand. Coeff. | t | Sig. |
| | B | Std. Error | Beta | | | B | Std. Error | Beta | | |
|---|---|---|---|---|---|---|---|---|---|---|
| Intercept: | 2.911 | 0.347 | | 8.393 | <0.001 * | 2.759 | 0.188 | | 14.671 | <0.001 * |
| M1F1: Aut. mot. astronomer | -0.018 | 0.041 | -0.016 | -0.435 | 0.664 | 0.04 | 0.03 | 0.039 | 1.364 | 0.173 |
| M1F2: Contr. mot. astronomer | -0.045 | 0.022 | -0.079 | -2.012 | 0.045 * | -0.038 | 0.016 | -0.065 | -2.312 | 0.021 * |
| M2F1: Aut. mot. publishing | 0.045 | 0.043 | 0.043 | 1.056 | 0.292 | -0.067 | 0.029 | -0.07 | -2.336 | 0.02 * |
| M2F2: Contr. mot. publishing | -0.003 | 0.038 | -0.003 | -0.074 | 0.941 | -0.037 | 0.028 | -0.042 | -1.285 | 0.199 |
| M3F1: Identified reg. publishing | 0.021 | 0.022 | 0.037 | 0.955 | 0.34 | -0.019 | 0.016 | -0.033 | -1.153 | 0.249 |
| M3F2: Introjected reg. publishing | 0.044 | 0.024 | 0.079 | 1.8 | 0.072 | 0.085 | 0.018 | 0.153 | 4.687 | <0.001 * |
| M3F3: External reg. publishing | 0.007 | 0.033 | 0.01 | 0.21 | 0.834 | 0.144 | 0.022 | 0.226 | 6.481 | <0.001 * |
| PPQ: Perceived Pub. Pressure | 0.262 | 0.041 | 0.304 | 6.45 | <0.001 * | *not included in Model 2* | | | | |
| DJ: Effort | 0.07 | 0.037 | 0.08 | 1.873 | 0.062 | *not included in Model 2* | | | | |
| DJ: Reward | -0.071 | 0.038 | -0.093 | -1.884 | 0.06 | *not included in Model 2* | | | | |
| OJ: Resource Allocation | 0.009 | 0.036 | 0.011 | 0.253 | 0.8 | *not included in Model 2* | | | | |
| OJ: Peer Review | -0.093 | 0.038 | -0.104 | -2.474 | 0.014 * | *not included in Model 2* | | | | |
| OJ: Grant Application | -0.031 | 0.036 | -0.037 | -0.879 | 0.38 | *not included in Model 2* | | | | |
| OJ: Telescope Application | -0.145 | 0.038 | -0.159 | -3.77 | <0.001 * | *not included in Model 2* | | | | |
| OC: Overcommitment | 0.004 | 0.034 | 0.005 | 0.107 | 0.915 | *not included in Model 2* | | | | |
| Gender: Male | 0.004 | 0.054 | 0.003 | 0.071 | 0.944 | -0.068 | 0.041 | -0.047 | -1.673 | 0.095 |
| Position: PhD Candidate | 0.147 | 0.135 | 0.043 | 1.092 | 0.275 | 0.159 | 0.073 | 0.081 | 2.188 | 0.029 * |
| Position: Postdoc | 0.079 | 0.073 | 0.051 | 1.084 | 0.279 | 0.14 | 0.053 | 0.092 | 2.62 | 0.009 * |
| Position: Assistant Prof. | 0.162 | 0.083 | 0.078 | 1.959 | 0.051 | 0.256 | 0.07 | 0.11 | 3.652 | <0.001 * |
| Position: Associate Prof. | 0.012 | 0.071 | 0.007 | 0.164 | 0.87 | -0.044 | 0.057 | -0.024 | -0.762 | 0.446 |
| Position: Other | -0.012 | 0.08 | -0.006 | -0.15 | 0.881 | -0.029 | 0.062 | -0.015 | -0.474 | 0.636 |
| Primary Employer: Academic | -0.15 | 0.092 | -0.06 | -1.627 | 0.104 | -0.114 | 0.075 | -0.042 | -1.514 | 0.13 |
| Papers published: Submission | *Excluded (filtered due to OJ: Peer Review)* | | | | | -0.324 | 0.168 | -0.056 | -1.926 | 0.049 |
| Papers published: 0 | *Excluded (filtered due to OJ: Peer Review)* | | | | | -0.188 | 0.149 | -0.035 | -1.266 | -0.079 |
| Papers published: 6–20 | -0.013 | 0.067 | -0.01 | -0.203 | 0.84 | -0.011 | 0.046 | -0.008 | -0.242 | 0.809 |
| Papers published: >20 | 0.041 | 0.065 | 0.031 | 0.626 | 0.531 | 0.052 | 0.049 | 0.037 | 1.064 | 0.288 |
| Location: Global North | 0.166 | 0.07 | 0.089 | 2.367 | 0.018 * | -0.022 | 0.049 | -0.013 | -0.462 | 0.644 |

* indicates statistical significance (p < 0.05).

biggest reasons to take on their profession are their intrinsic motivation to follow their curiosity about the universe and the intellectual challenge of studying astrophysical phenomena. Indeed, I find that curiosity and enjoyment of the intellectual challenge are crucial motivational factors for the respondents to become an astronomer and the comparatively biggest one is the "enjoyment of the process of gaining insight in astronomical phenomena". The findings indicate that autonomous motivation to become an astronomer dominates the controlled one (H1).

Consistent with the findings on motivational factors to become an astronomer, the respondents ranked as most rewarding about their work to be "enjoying the process of finding truths about the universe" and "making incremental & ground-breaking steps in building up knowledge". Moreover, the comparatively biggest driver to publish is the importance "to share

results with the community". These results are congruent with astronomer's autonomous motivation to push knowledge forward (cf. [14]); Astronomers enjoy (intrinsic motivation) to unravel the mysteries of the universe and find it important to share their results to push knowledge forward, while not enjoying "the process writing a paper" and the paper review process as much (comparatively smallest driver to publish). This means that their comparatively biggest driver to publish is one out of identified regulation–to get the results to the community for others to build on the knowledge. As hypothesised (H2), however, controlled autonomous dominates over autonomous motivation to publish. These controlled motivational factors mainly relate to the need of papers for one's career. Hence, while respondents rank "getting a paper published" only as a medium rewarding experience, the results show that it has significant importance for their career.

The importance of publishing for astronomers' career is emphasised by the fact that most respondents reported the need to boost their publication record for increasing their career chances as the biggest source for publication pressure. By contrast, a third less respondents reported that "only getting results out will push knowledge forward" sparks their publication pressure, even though this source of pressure would fit more with their autonomous motivation to push knowledge forward.

When analysing what role-associated and individual aspects play a role in the extent of the seven motivational factors I deducted from the factor analysis, I find that men feel a lower autonomous motivation and a higher controlled motivation for becoming an astronomer than females/ non-binaries. I can only speculate about the reasons: In a field that is still male-dominated, women may feel more that they have chosen astrophysics by their own accord, out of curiosity, as opposed to due to external factors. By contrast, the controlled motivation, as well as external, introjected and identified regulations to publish are found to be higher for females/ non-binaries in comparison to males. This supports hypothesis (H3), however only partly, since the autonomous motivation to publish also tends to be higher for females/ non-binaries than for males. The trend I find when looking at how the academic position of an astronomer influences the seven motivational factors also supports hypothesis (H3): higher academic positions are generally related to a higher autonomous motivation to publish. By contrast, the controlled motivation as well as the introjected and external regulations to publish are found to be lower for astronomers with a higher academic rank. This is consistent with the finding in [12] that those with a higher rank perceive less publication pressure compared to those with a lower rank. It makes sense that astronomers who feel more controlled than autonomous motivation to perform a certain action also feel more pressure to do so. Relatedly, astronomers who haven't published at all feel a higher interjected and external regulation to publish as compared to those who have published 1–5 papers in the last five years. The opposite is true for those who have published more than 20 papers in the last five years. Lastly, I found that astronomers employed by an institution in the Global North report less controlled motivation to become an astronomer and less autonomous motivation to publish. External regulation to publish, by contrast, is found higher as compared to those primarily employed in the Global South. Interestingly however, Heuritsch [12] found that astronomers employed in the Global North feel less publication pressure than those employed in the Global South. The reasons for the differences of Global North versus Global South deserve further investigation.

The comparison between motivational factors of becoming an astronomer and to publish leads us to conclude a disagreement in values: While the autonomous motivation to become an astronomer is bigger than the controlled one, the controlled motivation to publish is bigger than the autonomous one. This answers RQ1: No–the motivation for astronomers to perform research does not seem to be congruent with their motivation to publish papers. I therefore find quantitative support for the evaluation gap and anomie found in [14]–a discrepancy

between the importance of autonomous motivation to push knowledge forward and the controlled motivation to produce the most valued outcomes (publications).

Gagné et al. [17] report on the negative relation between the feeling of work autonomy and turnover intentions, which inspired hypothesis H4. To test this among academic astronomers may give an indication of whether this relation could also be valid in academia or only more commercial work environments. I found indeed that astronomers who report a higher autonomous motivation to become an astronomer and a higher autonomous motivation to publish contemplate less often to leave academia. In contrast, controlled motivation, as well as external and introjected regulation to publish increases the likelihood for an astronomer to consider leaving academia. As for primarily non-academic astronomers the results for the influence of motivational factors on turnover were not statistically significant. I also found support for hypothesis H5; a higher autonomous motivation to become an astronomer and a higher autonomous & identified motivation to publish increase the likelihood of astronomers to report that they love their job and the opposite is true for a higher introjected and external regulation to publish.

By testing H4 and H5, I answered RQ2. Yes, generally it seems to hold for academic astronomers, that a higher autonomous motivation to become an astronomer and to publish increases the love for their job and decrease their turnover intentions, while the opposite is the case with controlled types of motivation. Only a higher controlled motivation to become an astronomer also increases the chance that one is happy with their job.

When directly comparing the regression model with perceived publication pressure as a DV with the model by Heuritsch ([12]: Table 3 therein; N = 520), I notice that the gender and current position of the astronomer have lost its statistical significance. This may be party due the difference in N and partly due to the mediated effects of the added motivational factors. As evident from *section 4.3.*, lower ranked positions than Full Professor tend to feel a lower autonomous motivation to publish and a higher controlled motivation, introjected and external regulation to publish. The perception of publication pressure increases with introjected and external regulation to publish and decreases with autonomous motivation and identified regulation to publish, which supports hypothesis H6. Combining those two observations infer that astronomers who are lower ranked than Full Professor perceive more publication pressure, which is congruent with the results in [12]: p.15] observation, who didn't have motivational factors in their model. Motivational factors may therefore be a crucial mediator, as RCT suggests. The results for the perception of effort put into one's work are comparable (hypothesis H7): perceived effort increases when one feels a higher controlled motivation and introjected & external regulation to publish and decreases with a higher autonomous motivation to publish. The results for the perception of obtaining rewards from one's job are inconclusive: while a higher autonomous and controlled motivation to publish increase the likelihood to feel rewarded, a higher introjected and external regulation to publish lead astronomers to perceive their work less rewarding. I cannot explain this discrepancy between controlled motivation versus introjected and external regulation to publish, since the former by definition comprises the latter two concepts and–as I have shown–these concepts correlate positively, as expected. Therefore, hypothesis H8 is not supported. As for perceived overcommitment, hypothesis H9 is partly supported: taking into account that publication pressure, perceived effort & reward and procedural justice act as a mediator, a higher introjected and identified regulation to publish have direct positive effects on overcommitment. As for perception of frequency of misbehaviour, I find H10 partly supported: while I do find evidence for autonomous motivation to publish to decrease the perception of misconduct occurrence, and introjected and external regulation to publish to lead to an increase, I find that controlled motivation to become an astronomer decreases, instead of increases, this perception. When accounting for PPQ, ERI, OC and

Ojs as mediators (such as in the model in [12]), the only direct effect from a motivational factor that is statistically significant is controlled motivation to become an astronomer.

Therefore, with regards to RQ3, the answer is less straightforward as with the former two research questions. Yes, generally, I do observe autonomous types of motivation to lead to less publication pressure and higher cultural justice perceptions and the opposite for controlled types of motivations. However, there were also some surprising results, which would deserve further investigation in the future, such as the discrepancy of the effects of controlled motivation versus introjected and external regulation to publish on feeling rewarded for one's work or why a higher controlled motivation to become an astronomer decreases the perception of research misconduct.

## 6. Strengths/Limitations

A detailed analysis of the strengths and limitations (such as response bias) of the web-based survey this study is based on can be found in [12]. For this study specifically, I point out that the Cronbach Alphas for the motivation constructs could be improved by enhancing the item batteries. Especially for M3 there is only one item drawing on identified regulation. Despite extensive search in literature, I found it challenging to design more items representing this kind of motivation for the question "How do you feel when you don't publish the amount of papers that you aimed to publish?".

Furthermore, I observe slight differences in the regression results compared to the study by Heuritsch [12]. These may stem from the fact, that for the analysis in this study I used SPSS's linear regression algorithm, instead of specifying and calculating the whole SEM model with Lavaan in R (leading differences in N). I chose this simpler method as adding the additional seven motivational constructs into the SEM would make the model too complex and I would risk overspecification.

## 7. Conclusion, implications & outlook for further research

The aim of this research was to study the motivations of people becoming an astronomer and to publish as an output of their work and their effects on astronomer's research behaviour. I distinguish between autonomous, arising of the internal component of an actor's situation, and controlled types of motivation, arising of the external components of the actor's situation–the culture, norms and material opportunities present at action. I found a discrepancy between the importance of publications as a personal reward and their importance for astronomer's career. In other words, while astronomers' biggest autonomous goal is to push knowledge forward (cf. [14]), their biggest drivers to publish are related to their career prospects. Hence, what performance indicators, such as the publication rate, measure diverges from what astronomers value. This evaluation gap could indeed play into the balance act found in [14], where astronomers aim at producing quality science that pushes knowledge forward, while at the same time sacrificing just as much quality that is necessary to also produce the expected quantity of papers.

Recent research (e.g. [17–19]) on organisational psychology shows that autonomous motivation is related to better work outcomes (e.g. performance, well-being, competence etc) and indeed I find that a higher autonomous motivation to publish decreases perception of publication pressure while a higher introjected & external regulation to publish increase this perceived pressure. The same goes for perceived effort put into work. As the results in [12] show, perceived publication pressure, in turn, increases the perception of a higher scientific misconduct. I conclude that driving up factors of controlled motivation through incentives based on performance indicators, such as publication rate, may have negative effects on astronomer's well-

being and research conduct. Moreover, while identified regulation to publish increases over-commitment, it decreases publication pressure and increases how much astronomers like their job. As Gagné et al. [18] point out, studies have shown that performance was more highly correlated with identified than with intrinsic motivation. Moreover, I found that autonomous types of motivation are positively related to astronomers loving their jobs, while controlled types of motivations are positively related to turnover intentions. Hence, I suggest that the promotion of the internalisation of the value of the task (such as publication) would be beneficial for research in astronomy. In order for that value to be internalised however, the way how publications are done may need to be adapted for them to better fit the quality requirements of astronomers (cf. [14]).

Therefore, I draw two main conclusions:

1. We found quantitative evidence for an evaluation gap, where what is valued in academia (producing papers) is incongruent with the astronomer's autonomous motivational factors to conduct research. Having to attain a goal (such as writing papers to survive in academia) which is not internalised, drives up astronomers' controlled motivation.

2. Driving up factors of controlled motivation through incentives, such as the publication rate, in turn, were found to have negative effects on astronomer's perceptions of work environment justice and research conduct.

Future studies could inquire, together with astronomers, what innovative publications, which are better aligned with the astronomers' quality criteria, could entail. Moreover, further research may uncover more aspects of research that would relate to astronomers' autonomous motivation and suggest how evaluation criteria could be designed on that basis. Building upon the results of that kind of research, could inspire incentives which leverage astronomers' autonomous motivation, and thereby increasing their well-being and work engagement. After all, "participation in planning and evaluation was [found to be] related to satisfaction, while participation in planning was related only to productivity" [24: p.173].

## Supporting information

**S1 Fig.**
(JPG)

**S1 Appendix. Survey questions.**
(DOCX)

**S2 Appendix. EFAs and CFAs.**
(DOCX)

**S3 Appendix. Results.**
(DOCX)

**S1 File. Survey data.**
(ZIP)

## Acknowledgments

First, I would like to extend my gratitude to the 3509 astronomers who dedicated upwards of half an hour—despite the publish-or-perish imperative—to participate in this survey. Second, Thea Gronemeier and Florian Beng assisted the survey design and were a big support in data processing. Third, I would like to thank my supervisor, Stephan Gauch, for facilitating this

project. Fourth, thank you to all the pre-testers: Niels Taubert, Jens Ambrasat, Andrej Dvornik, Iva Laginja, Levente Borvák, Nathalie Schwichtenberg, Theresa Velden, Richard Heidler, Rudolf Albrecht, Andreas Herdin, Alex Fenton and Philipp Löschl. Fifth, I would like to thank Marion Bräuer, whose graphic design expertise flowed into the visualisation of the motivation continuum and Nanou Haafs-Baart, who advised me on making the text more readable. Finally, I am grateful to GESIS (Lebniz Institut für Sozialwissenschaften) for the scholarship that enabled me to participate in their 2019 survey methodology course and the support by the Open Access Publication Fund of Humboldt-Universität zu Berlin.

## Author Contributions

**Conceptualization:** Julia Heuritsch.

**Data curation:** Julia Heuritsch.

**Formal analysis:** Julia Heuritsch.

**Investigation:** Julia Heuritsch.

**Methodology:** Julia Heuritsch.

**Visualization:** Julia Heuritsch.

**Writing – original draft:** Julia Heuritsch.

**Writing – review & editing:** Julia Heuritsch.

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
