## [Decision Letter · Decision Letter 0]

12 Jul 2022

PONE-D-22-10940Reflecting on Motivations: How Reasons to Publish affect Research Behaviour in AstronomyPLOS ONE

Dear Dr. Heuritsch,

Thank you for submitting your manuscript to PLOS ONE. After careful consideration, we feel that it has merit but does not fully meet PLOS ONE’s publication criteria as it currently stands. Therefore, we invite you to submit a revised version of the manuscript that addresses the points raised during the review process.

As you can see from the reports, the main issues relate the structure and, generally, the clarity of the paper. Both reviewers recommend substantial shortening, but also giving it a clear, easy to understand structure. This does not imply, of course, suppressing some of the results, but rather focusing in the paper on the main hypotheses testing, and moving all other possible (exploratory) analyses to the appendix. It means also starting out with a clear statement of the research questions, and then following through.

There are two important methodological points: Reviewer 1 points out that it seems the data is the same as for another paper - if this is true, be completely open about the relationship between this and the other paper. If it is not true, make sure that it doesn´t seem to be the case!

Secondly, reviewer 1 points out that it seems that the EFA and CFA have been run on the same dataset. Again, if this is not true, make sure it does not appear that way. If it is true, you should reconsider (and reviewer 1 gives some literature about it). If you cannot change it, you have to discuss it carefully! Please have a look at the detailed points made by both reviewers, especially with respect to the exposition of the paper, and use them as guidance to improve the paper.

We look forward to receiving your revised manuscript.

Kind regards,

Christiane Schwieren, Dr.

Academic Editor

PLOS ONE

Journal Requirements:

3. For this single-authored manuscript, please replace "we" with "I".

*Please change "female” or "male" to "woman” or "man" as appropriate, when used as a noun (see for instance https://apastyle.apa.org/style-grammar-guidelines/bias-free-language/gender).

Reviewers' comments:

Reviewer's Responses to Questions

**Comments to the Author**

1. Is the manuscript technically sound, and do the data support the conclusions?

Reviewer #1: Yes

Reviewer #2: Yes

2. Has the statistical analysis been performed appropriately and rigorously? 

Reviewer #1: I Don't Know

Reviewer #2: I Don't Know

3. Have the authors made all data underlying the findings in their manuscript fully available?

Reviewer #1: No

Reviewer #2: Yes

4. Is the manuscript presented in an intelligible fashion and written in standard English?

Reviewer #1: Yes

Reviewer #2: Yes

5. Review Comments to the Author

Reviewer #1: The article describes a large survey conducted among astronomers to understand how their motivations play a role in publishing and research behaviour. The topic is timely and the insights are useful, but although PLOS ONE does not dictate word limits, I think the piece would be more useful when written up more concisely. I have a few suggestions that I hope to be useful for the author in revising the piece:

General remarks

It may help the author to think a bit more about how the various hypotheses help answer the main research question, and to pay more attention to those (i.e., research questions) in the Introduction, Results and Discussion section. This may mean that substantial number of Tables should be moved to the supplementary information, so that the information is available to interested readers, but that general readers get a bit more support from the author in piecing together the different models.

I also wanted to note that the author says all data is available, but I saw no link to a public repository or a raw dataset anywhere. This may be due to the type of consent participants have provided, but then the author should explain this.

Please note that the complete review can be found attached.

Reviewer #2: Dear author,

Thank you for having the opportunity to read and review with manuscript which I did with joy and pleasure. Great work and great way to further dive into the motivations of researchers. A couple of suggestions belolw that may improve the manuscript:

Abstract:

Please assure that others also know what is meant with reflexive metrics. I know the author is very experienced with the term but PLOS One readers may not and should be taken by hand :)

Please make sure that readers know what anomie is

Now the abstract does not contain any methods and I think this should be captures in more detail in the abstract. What is done? What Questionnaires are used, what types of participants, what hypothesis etc. I think the IMRaD format can help here.

Introduction:

How is resaerch culture shaping resaerch behaviour. Is there evidence for this?

And how is what types of behaviour shaped by what elements of culture?

Is 10% of the variance a reference to the work of Heuritsch 2021?

Maybe elaborate on what the author means with institutional norms? They sound vague and not so concrete. More specific? What is considered these norms?

It is a bit weird to talk about we, while it is a single author paper.

Nice to see how the author elaborates on how the paper is structured!

Are there any RQs worth mentioning in the introduction? IOW: why this paper? What is the urgency?

Methods:

Matter of style, but response rate etc is oftentimes reported in the results section.

How is the estimation of 13000-15000 calculated. It sounds a bit too far off and it is a bit weird to calculate a response rate, while you dont know how many people are reasched. Better to hightlihgt this limitation in discussion section, which you most likely do.

Instruments:

Are the 18 items in the supplementary material? How are these 18 items chosen and why?

Make sure that all the instruments are supplemented in these sections.

What was the result fo the factor analysis of the M1 instrument? Please report this in the paper

Motication to publish: what did you do now and why? the paragraph is a bit unclear. Is there any psychometric validation? What constructs and concepts are measured?

What are the ethical considerations and assessment by an ERB?

Is there a privacy policy?

Is there informed consent?

Is there a preregistration with these hypothesis somewhere available?

Results:

I feel that I am a bit lost in all the tables and details. What does the author want to say? What are the key findings? What is the main focus of the paper? Is there a content that should be focused on.

I suggest to highlight the most important data in 3 tables and move the rest to the supplementary materials

Discussion:

The limitations section is very short to put it mildly. What are the strengths of the paper?

and what are the limitations (response rate, response bias, confirmation bias? Reporting bias? No valid Qs? no generalisation? Comparison with other countries about levels?

What is your main conclusion after all these data that you have collected? What are the main questionnaires that are used that give the most valuable information? Now it feels a bit as swimming in a swimming pool with too many data

6. PLOS authors have the option to publish the peer review history of their article (what does this mean?). If published, this will include your full peer review and any attached files.

Reviewer #1: No

Reviewer #2: No

---

## [Author Response · Author response to Decision Letter 0]

26 Sep 2022

The full response to both reviewers and the editor can be found in "Response to Reviewers.docx".

---

## [Decision Letter · Decision Letter 1]

27 Jan 2023

Reflecting on Motivations: How Reasons to Publish affect Research Behaviour in Astronomy

PONE-D-22-10940R1

Dear Dr. Heuritsch,

We’re pleased to inform you that your manuscript has been judged scientifically suitable for publication and will be formally accepted for publication once it meets all outstanding technical requirements.

Kind regards,

Christiane Schwieren, Dr.

Academic Editor

PLOS ONE

Additional Editor Comments (optional):

Reviewers' comments:

Reviewer's Responses to Questions

**Comments to the Author**

1. If the authors have adequately addressed your comments raised in a previous round of review and you feel that this manuscript is now acceptable for publication, you may indicate that here to bypass the “Comments to the Author” section, enter your conflict of interest statement in the “Confidential to Editor” section, and submit your "Accept" recommendation.

Reviewer #2: All comments have been addressed

2. Is the manuscript technically sound, and do the data support the conclusions?

Reviewer #2: Yes

3. Has the statistical analysis been performed appropriately and rigorously? 

Reviewer #2: I Don't Know

4. Have the authors made all data underlying the findings in their manuscript fully available?

Reviewer #2: Yes

5. Is the manuscript presented in an intelligible fashion and written in standard English?

Reviewer #2: No

6. Review Comments to the Author

Reviewer #2: (No Response)

7. PLOS authors have the option to publish the peer review history of their article (what does this mean?). If published, this will include your full peer review and any attached files.

Reviewer #2: No

---

## [Editor Report · Acceptance letter]

8 Feb 2023

PONE-D-22-10940R1 

Reflecting on motivations: How reasons to publish affect research behaviour in astronomy 

Dear Dr. Heuritsch:

I'm pleased to inform you that your manuscript has been deemed suitable for publication in PLOS ONE. Congratulations! Your manuscript is now with our production department. 

Kind regards, 

on behalf of

Dr. Christiane Schwieren 

Academic Editor

PLOS ONE